**Investigation**

# Posterior estimation of longitudinal variance components from nonlongitudinal data using Bayesian Gaussian process model

Arttu Arjas [ID],[1,*] Kalle Leppälä [ID],[2] Mikko J. Sillanpää [ID] [3]

[1]Centre for Wireless Communications, University of Oulu, Oulu 90014, Finland
[2]The Organismal and Evolutionary Biology Research Programme, University of Helsinki, Helsinki 00014, Finland
[3]Research Unit of Mathematical Sciences, University of Oulu, Oulu 90014, Finland

*Corresponding author: Centre for Wireless Communications, University of Oulu, Oulu 90014, Finland. Email: arttu.arjas@oulu.fi

Many quantitative traits can be measured from a single individual only once, making acquisition of longitudinal data impossible. In this paper, we present Gaussian process restricted Bayesian estimation, a new method tailored for estimating posterior distributions of longitudinal variance components from data where each individual contributes only 1 measurement at a single time point to the study. However, by collecting all time points together, one can think data to be longitudinal at the population level which makes it possible to estimate longitudinal variance components. The method can be also applied for reaction norm problems where it is common that a value of continuous environmental condition (e.g. temperature) is measured only once per individual. The work is based on Bayesian framework, Markov chain Monte Carlo estimation, and assuming Gaussian process-based smoothing priors for the variance components. The performance of the method is illustrated with simulated and real data sets as well as compared with a random regression model. Our method is very stable and it is flexible in handling any kind of smooth curves. Uncertainty around the variance curves is represented with 95% credible interval curves computed from the posterior distribution. The code is available at the GitHub repository https://github.com/aarjas/GP-REBE.

**Keywords:** Gaussian process; dynamic heritability; MALA; REML

## Introduction

Expression of many traits depend on time, age, and environmental exposures or conditions. Thus, it is natural to study such biological process traits (i.e. character processes or function-valued traits) in nature by measuring them longitudinally over time or over different conditions (Stinchcombe and Kirkpatrick 2012). Combining measurements together and analyzing them jointly as a single function-valued trait will often provide more power and create new biological insights. This, however, requires new specialized analysis methods which are fortunately rapidly emerging to the field (Moore et al. 2013; Li and Sillanpää 2015).

Measuring some breeding traits in animals like carcass traits requires killing the animal. Similarly, measuring certain wood quality traits in trees and detecting frost in plants may require destroying the particular tree or a plant. This makes it impossible to collect longitudinal data in the usual sense. The setting where a single individual contributes only a single data point will be referred to as group-level longitudinal, when the data as a whole cover a wide time span. A closely related case to group-level longitudinality also arises naturally when we instead of time consider some other continuous covariate such as temperature, latitude, or some trait of the individual other than the one under study. Here, the continuous covariate acts as time; each individual has 1 measurement of the covariate just like each individual is represented in

1 time point in group-level longitudinal data. Many traits exhibit phenotypic plasticity, i.e. expression sensitive to environmental context due to interaction between genetic and environmental factors. The interaction of genotypes with a continuous covariate is described by the reaction norm (Gregorius and Namkoong 1986; Jarquín et al. 2014), which also exhibits group-level longitudinality.

Many of the analysis frameworks for estimating dynamic heritability (the proportion of trait variation attributable to genetic factors) have been developed for settings where a single trait has been measured from a single individual at several consecutive time points—data are longitudinal at individual-level (Arjas et al. 2020). However, there are only few methods, in particular random regression models (RRMs), developed for a setting where a single individual is measured only at a single time point but the individuals at population-level (large population-based sample) can be considered jointly to contribute to the longitudinal sample (Schaeffer 2016; Robinson et al. 2017; Moore et al. 2019; Ni et al. 2019). In other words, individuals of the sample can be arranged to the continuum of age-groups all the way from young individuals to very old ones so that they together will cover several time points—data are longitudinal at group-level.

Using RRMs, some recent works are able to consider gene-covariate interactions with possibly only 1 measurement per individual (Moore et al. 2019; Ni et al. 2019). Theoretically, to perform such analysis, the covariate would have to be split up into discrete

groups with each group containing suitably large population (Robinson *et al.* 2017). This would also result in each group having their own relationship matrix. In these studies, for simplicity, only intercept, slope, and quadratic terms were considered to model breeding value curves, which imply slightly more complicated form for the dynamic variance components. In general, more complicated functions (e.g. Legendre polynomials or splines) are still possible to use with RRMs.

Jaffrézic and Pletcher (2000) compared 3 different and closely related approaches to model variances (and heritabilities) of function-valued traits. The 3 approaches are: RRMs, orthogonal polynomial approximation, and character process model. In orthogonal polynomial and character process models, covariance functions are approximated either with orthogonal polynomials or with parametric functions which are able to reduce the total number of model parameters. In principle, all of these may be with slight modifications applicable to our setting but we consider the RRM as our main competitor here because it seems to be the most commonly used approach in practice.

In this work, we present a Bayesian Gaussian process (GP) model for estimating dynamic variance components for group-level longitudinal data. Our model is very flexible with respect to functional form of variance curves, not requiring to choose a basis. Unlike RRM, which could be unstable in case of having more than intercept and slope parameters, our approach provides very stable performance. The unstability of polynomial RRMs has been reported before by Robbins *et al.* (2005) and was noticed in our experiments as well. Our solution is fully Bayesian which allows posterior uncertainty quantification.

This paper is organized as follows: first we introduce the static and dynamic versions of the linear mixed-effects model. We then introduce a GP smoothness prior for the variance components. After that, we propose a Bayesian version of restricted maximum likelihood (REML) coupled with Metropolis-adjusted Langevin algorithm (MALA) for efficient estimation of the variance curves. We then discuss implementation details, analyses of simulated and real data, and present results together with comparing our method to RRM. We finish with a discussion of results and observations.

# Materials and methods
## Model description
### Static linear mixed-effects model

We start by considering a static version of the linear mixed-effects model which has been extensively used in quantitative genetics. The model for a quantitative trait $y \in \mathbb{R}^n$ is often written as

$$y = X\beta + Zu + e, \tag{1}$$

where $\beta \in \mathbb{R}^p$ denotes the fixed effects and $u \in \mathbb{R}^m$ the additive genetic values (random effects). The design matrices $X \in \mathbb{R}^{n \times p}$ and $Z \in \mathbb{R}^{n \times m}$ encode information about known covariates, grouping of measurements, etc. Residuals $e$ are independent of $u$. To facilitate estimation of genetic values $u$, there should be a priori information about the genetic relationship of the individuals. This can be inferred for example from a pedigree or genomic data and coded into a covariance matrix, which is then called either pedigree-based additive genetic relationship matrix (Mrode and Thompson 2005) or genomic additive relationship matrix (VanRaden 2008). Setting $Z = I$, the genetic effects are then assumed a Gaussian distribution, i.e. $u \sim \mathcal{N}(0, \sigma_G^2 G)$, where $G \in \mathbb{R}^{n \times n}$

is the relationship matrix and $\sigma_G^2$ is the additive genetic variance component. The residuals $e$ are also assumed to follow a Gaussian distribution, $e \sim \mathcal{N}(0, \sigma_E^2 I)$.

### Proposed dynamic linear mixed-effects model

If we now assume that a single individual contributes only a single phenotypic measurement to the vector $y$, this yields the model (1) but with dynamic variance components. In this dynamic case, we let the variance components vary as a function of some continuous covariate, for instance time or temperature. The model is written as (cf. (1))

$$\tilde{y} = X\beta + Z\tilde{u} + \tilde{e}. \tag{2}$$

The dimensions of the variables are the same as in Equation (1). We use tilde ($\tilde{\ }$) to distinguish dynamic variables from constant ones; $\tilde{u}$ and $\tilde{e}$ are still independent. We denote by $\sigma_G^2(t_i)$ and $\sigma_E^2(t_i)$ the value of the genetic and residual variance components at point $t_i$ (the measured value of the covariate on the $i$th individual). We then introduce diagonal matrices $D_G \in \mathbb{R}^{n \times n}$ and $D_E \in \mathbb{R}^{n \times n}$ and set $(D_G)_{ii} = \sigma_G(t_i)$ and $(D_E)_{ii} = \sigma_E(t_i)$. We assume that $\tilde{u} \sim \mathcal{N}(0, D_G G D_G)$ and $\tilde{e} \sim \mathcal{N}(0, D_E^2)$, which implies $\mathrm{var}(\tilde{u}_i) = \sigma_G^2(t_i) G_{ii}$, $\mathrm{cov}(\tilde{u}_i, \tilde{u}_j) = \sigma_G(t_i)\sigma_G(t_j) G_{ij}$ and $\mathrm{var}(\tilde{e}_i) = \sigma_E^2(t_i)$, $\mathrm{cov}(\tilde{e}_i, \tilde{e}_j) = 0$. The (narrow-sense) heritability at point $t_i$ is then given by the formula $h^2(t_i) = \sigma_G^2(t_i)/(\sigma_E^2(t_i) + \sigma_G^2(t_i))$. We point out that even though the matrices are diagonal, we may enforce some smoothness in the variance curves through the Matérn covariance which models the dependencies between time points, as explained below. Moreover, we note that this model is an extension of the model presented in Arjas *et al.* (2020). The previous work assumed that all $n$ individuals were measured at $t$ common, equidistant time points, yielding a total of $nt$ measurements. In this work, we assume that there are in total $n$ measurements (1 measurement per individual) taken at arbitrary time/covariate points.

### Prior distribution of the dynamic variance components

In our model, the variance components are essentially functions of some covariate like time or temperature. It is natural to expect some sort of regularity of the functions, i.e. continuity and smoothness. These assumptions can be enforced with GPs [see Williams and Rasmussen (2006) for introduction to the topic]. Essentially, GPs are continuous-time stochastic processes, where for any finite-dimensional index set, the process follows a multivariate normal distribution. A GP is entirely defined by a mean function and a covariance function. Regularity of the process is enforced through the covariance function. A typical choice is the Matérn covariance function

$$C(t, t'; \tau^2, \nu, \ell) = \tau^2 \frac{2^{1-\nu}}{\Gamma(\nu)} \left( \frac{\sqrt{2\nu}|t - t'|}{\ell} \right)^\nu K_\nu \left( \frac{\sqrt{2\nu}|t - t'|}{\ell} \right) \tag{3}$$

between points $t$ and $t'$, where $K_\nu$ is the modified Bessel function of the second kind of order $\nu$, $\tau^2$ is the magnitude, $\nu$ is the smoothness, and $\ell$ is the length-scale of the process. The smoothness $\nu$ controls the differentiability of the process, i.e. the process is $k$ times differentiable in the mean-square sense if $\nu > k$. The magnitude $\tau^2$ controls overall variation of the process and length-scale $\ell$ controls variations in a smaller scale.

We use a GP with Matérn covariance function to model the variance component functions. This involves discretization of the covariate domain (see "*Computational grid*" section), and modeling

the variance curves as multivariate Gaussians in these discretization points. The covariance matrices of the Gaussians are computed at each pair of points with the Matérn covariance function. Since GP does not constrain the variables and we expect variances to be nonnegative, we employ the transform $s_{G/E}(t_i) = \log(\sigma_{G/E}^2(t_i))$ and model the transformed variable with a GP instead. That way the variances are guaranteed to be positive after inverting the transformation.

## Parameter estimation

### Restricted Bayesian variance estimation

For estimation of the dynamic variance curves in model (2), we apply similar transformation as in REML estimation (Patterson and Thompson 1971) in the Bayesian framework. This allows us to estimate variance components independently from the fixed effects. Setting $K = D_G G D_G + D_E^2$, the restricted log-likelihood function of the data given the transformed variance components $s = [s_E^\mathsf{T}, s_G^\mathsf{T}]^\mathsf{T}$ is given by

$$\ell_R(\tilde{y}\,|\,s) = \text{const.} - \frac{1}{2}\log|K| - \frac{1}{2}\log|X^\mathsf{T} K^{-1} X| - \frac{1}{2}\tilde{y}^\mathsf{T} P \tilde{y}, \qquad (4)$$

where $P = K^{-1} - K^{-1} X (X^\mathsf{T} K^{-1} X)^{-1} X^\mathsf{T} K^{-1}$ (Searle *et al.* 1992; Johnson and Thompson 1995). Bayes' formula is then utilized to add the smoothing effect of the GP priors. This yields the unnormalized log-posterior density of the logarithmized variance components

$$\log \pi(s\,|\,\tilde{y}) = \text{const.} + \ell_R(\tilde{y}\,|\,s) - \frac{1}{2}s_E^\mathsf{T} C^{-1} s_E - \frac{1}{2}s_G^\mathsf{T} C^{-1} s_G, \qquad (5)$$

where $C$ is a matrix of Matérn covariances evaluated at each pair of grid points (see "*Computational grid*" section for implementation details).

### Metropolis-adjusted Langevin algorithm

To explore the posterior distribution of transformed variance components $s$, we use MALA (Roberts and Rosenthal 1998; Girolami and Calderhead 2011). It uses Langevin dynamics to propose new values to a point cloud that approximates the posterior distribution of the variance curves. The values are either accepted or discarded based on an acceptance probability. Langevin diffusion for a density $\pi$ is defined as

$$\frac{ds}{dt} = \nabla \log \pi(s\,|\,\tilde{y}) + \sqrt{2}\frac{dW}{dt},$$

where $\nabla$ is the gradient with respect to $s$ and $W$ denotes standard Brownian motion. Approximating the Langevin diffusion in a discrete setting, the proposal for MALA at point $s_k$ with step size $\tau$ is given as

$$s_{k+1}^p = s_k + \tau A \nabla \log \pi(s_k\,|\,\tilde{y}) + \sqrt{2\tau A}\varepsilon_k,$$

where $\varepsilon_k$ follows standard multivariate Gaussian distribution and $A$ is a positive definite preconditioning matrix. To make sure that the algorithm converges to the correct target density, a Metropolis-Hastings step must be added. It involves computing an acceptance probability

$$\alpha_k = \min\left\{1, \frac{\pi(s_{k+1}^p\,|\,\tilde{y})q(s_k\,|\,s_{k+1}^p)}{\pi(s_k\,|\,\tilde{y})q(s_{k+1}^p\,|\,s_k)}\right\}, \qquad (6)$$

where $q(z\,|\,z') = \mathcal{N}(z\,|\,z' + \tau A \nabla \log \pi(z'), 2\tau A)$. The proposed point is then accepted with probability $\alpha_k$, otherwise the current iterate is chosen as the new iterate. When a large enough sample from the posterior distribution has been generated, it can be used for Bayesian point estimation and uncertainty quantification for variance component and narrow-sense heritability curves.

## Random regression models

RRMs were first introduced by Henderson (1982) and a tutorial for using them is presented in Schaeffer (2016). They are widely used in genetics and breeding community and will serve as a comparison point for our method. RRMs are linear mixed-effects models where the random effects are modeled with regressions with respect to some measured covariate, for example, time. A standard RRM for measurement $y \in \mathbb{R}^n$ can be written as

$$y = X\beta + \Phi_G a + e, \qquad (7)$$

where $X$, $\beta$, and $e$ are as in Equation (1). The difference to a basic linear mixed-effects model is in the random effects that are defined as $u = \Phi_G a$, where $\Phi_G \in \mathbb{R}^{n \times nq}$ is a basis function matrix and $a \in \mathbb{R}^{nq}$ is a vector of random regression coefficients. Typical bases include for example polynomials or splines. For cases with 1 measurement per individual, the basis matrix $\Phi_G$ is constructed such that at row $i$, columns $i$, $i+1$, ..., $i+q-1$ obtain the basis function values for individual $i$. Rest of the matrix entries are zero. This way, each individual has $q$ unknown random regression coefficients. The coefficients of each individual are stacked in the vector consecutively. The coefficient vector is assumed a Gaussian distribution $a \sim \mathcal{N}(0, G \otimes \Sigma_G)$, i.e. coefficients between individuals have covariance $G$ (genomic additive relationship matrix), and coefficients within individuals have covariance $\Sigma_G$. The covariance matrix of $\tilde{u}$ is thus given as $\text{cov}(\tilde{u}) = \Phi_G(G \otimes \Sigma_G)\Phi_G^\mathsf{T}$. To fit the model, REML estimation is used to find an estimate for $\Sigma_G$. The diagonal entries, $\text{diag}(\Phi_G(G \otimes \Sigma_G)\Phi_G^\mathsf{T}) = [\text{var}(u_1), \ldots, \text{var}(u_n)]^\mathsf{T}$, represent the genetic variances at each measured covariate point. Similar basis expansion can be used for the residual $e$ as well, in which case the relationship matrix $G$ is substituted with an identity matrix. The differences between GP restricted Bayesian estimation (GP-REBE) and RRM are illustrated in graph format in Fig. 1. As a final note, when fitting a RRM with first-order polynomials, the resulting variance curves will be of second order, resulting from computing the variances as diagonal values of $\Phi_G(G \otimes \Sigma_G)\Phi_G^\mathsf{T}$. In general, the variance curves will always be of higher order than the polynomial used to model the random effects.

## Implementation

### Computational grid

In the computations, one needs to discretize the continuous GP. This can be done by specifying a computational grid that spans the observed range of the covariate (e.g. time, age, or temperature) under consideration. The covariance function is then evaluated at each pair of grid points to produce the covariance matrix. To avoid numerical difficulties, the grid has to be sparse enough such that the condition number of the covariance matrix stays sufficiently low. To evaluate the likelihood, the GP must be interpolated to the measured covariate values. Let $[t_1, \ldots, t_N]^\mathsf{T} \in \mathbb{R}^N$ denote the grid. Interpolating the GP to an arbitrary point $t^* \in \mathbb{R}$ using known function values $f_g = [f(t_1), \ldots, f(t_N)]^\mathsf{T} \in \mathbb{R}^N$ at the grid points can be done as follows: let $C_g \in \mathbb{R}^{N \times N}$ be a covariance matrix of the GP evaluated at the grid points and $c^* \in \mathbb{R}^N$ a vector of pairwise

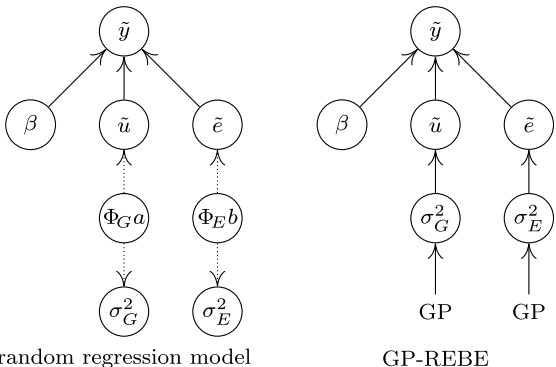

**Fig. 1.** Illustration of the differences between the RRM and our proposed model (GP-REBE). In random regression, the random effects $\tilde{u}$ are expressed in terms of basis functions $\Phi_G$ and unknown coefficients $a$. REML is used to estimate the covariance matrix of $a$, which is then used to form the covariance matrix of $\tilde{u}$. In our proposed model, the variance component functions are estimated directly with a Bayesian approach using a smoothing GP prior.

covariances between $t^*$ and the grid points. Interpolation is performed with a linear transform $f(t^*) = c^{*\top} C_g^{-1} f_g$. The unknown parameters of the model that are sampled with the MALA algorithm are the values of the variance component functions at the grid points. The number of grid points essentially controls the number of parameters needed to be estimated. In our analyses, we used $N = 50$ uniformly spaced grid points. We point out that the interpolation process might cause slight loss of accuracy in the estimates, depending on the distance of the grid points from the measured covariate values. This could be avoided by including the measured covariate values in the grid. However, this would increase the number of unknown parameters in the model by $2n$ and cause other computational issues that way. Moreover, if there are no large "gaps", i.e. areas with little or no measurements, in the measured covariate, the loss of accuracy is quite small. One could also consider nonuniform grids, where the grid values are chosen close to the measured covariate values. However, we leave this extension to future studies.

## Preconditioning

The convergence of MALA algorithm depends strongly on the choice of the preconditioning matrix $A$ (see "*Metropolis-adjusted Langevin algorithm*" section). A good choice for the matrix would be the inverse of Hessian matrix of the logarithmic posterior density evaluated at the current iterate since it carries information about the curvature of the density (Girolami and Calderhead 2011). In linear mixed-effects model literature, however, the Hessian of the restricted likelihood function is often replaced by the average information (AI) matrix, which is the average of Hessian and Fisher information matrices (Gilmour *et al.* 1995). This is because some computationally difficult terms cancel out when taking the average. Specifically, the element at $i$th row and $j$th column of the AI matrix with respect to the transformed variance components $s = [s_E^\top, s_G^\top]^\top$ is given as

$$
\begin{aligned}
\mathrm{AI}(s)_{i,j} &= -\frac{1}{2}\left\{ \frac{\partial^2 \ell_R}{\partial s_i \partial s_j} + \mathbb{E}\left( \frac{\partial^2 \ell_R}{\partial s_i \partial s_j} \right) \right\} \\
&= -\frac{1}{2} \tilde{y}^\top P \frac{\partial K}{\partial s_i} P \frac{\partial K}{\partial s_j} P \tilde{y}
\end{aligned}
\tag{8}
$$

(Johnson and Thompson 1995). In the Bayesian framework, we also need to take into account the prior density of variance

components in the preconditioning. Since the Hessian of the log-prior is simple to compute, we add that to the AI matrix. The (constant) Hessian of the log-prior is

$$
H_p = -I_2 \otimes C^{-1} = -\begin{bmatrix} C^{-1} & 0 \\ 0 & C^{-1} \end{bmatrix}.
\tag{9}
$$

The Kronecker product ($\otimes$) with identity matrix copies $C^{-1}$ twice to produce a block diagonal matrix. The matrix is block diagonal because we have assumed the additive genetic and residual variance components to be mutually independent a priori. The Bayesian AI (BAI) matrix is then $\mathrm{BAI}(s) = \mathrm{AI}(s) + H_p$. For additional computational benefits, we use a global approximation of the BAI matrix. This is to avoid computing the matrix, its square root, and inverse at every iteration of the MALA algorithm. For the global approximation, we first find the maximum a posteriori (MAP) estimate of transformed variance components using a Newton-Raphson type iterative method. We replace the Hessian (which is used in Newton-Raphson) with the BAI matrix. The update rule of the method is given as

$$
s_{k+1} = s_k + \mathrm{BAI}(s_k)^{-1} \nabla \log \pi(s_k \mid \tilde{y}).
\tag{10}
$$

The globally approximative BAI matrix is then computed at the MAP estimate. In our experiments, we found this global preconditioning matrix to perform very well, i.e. the Markov chains converged in a few thousand iterations.

## Hyperparameters of the GP

The Matérn covariance function has 3 hyperparameters that need to be chosen prior to estimating the additive genetic and residual variance component curves. In addition, we need to choose the mean function of the GP. We choose the mean function to be simply 0, and scale the data by dividing them with their sample standard deviation times $\sqrt{0.5}$, forcing the phenotypic variance of the data to be 2. This means that the genetic and residual variance components would be on average one, given that heritability of the trait is 0.5. After logarithmizing, the average is zero. Furthermore, we choose the magnitude $\tau^2$ of the GP to be 1. This means that 95% of the prior mass of logarithmized variances lies between $-1.96$ and $1.96$, and 95% of the prior mass of variances on their original scale lies between 0.14 and 7.1. We choose the length-scale $\ell$ to be equal to the distance between the highest and lowest covariate values. This yields quite regular curves but still allows some local variation. We point out that the flexibility of estimated curves depends on $\ell$. If one wants to recover very straight curves, $\ell$ should be set to a high value with respect to the covariate. Finally, we choose the smoothness $\nu$ to be 1.5. This is a standard choice as this causes the GP to be once mean-square differentiable a priori and allows the Matérn covariance function to be expressed in terms of elementary functions. The covariance function for $\nu = 1.5$ is

$$
C(t, t')_{1.5} = \tau^2 \left( 1 + \frac{\sqrt{3}|t - t'|}{\ell} \right) \exp\left( -\frac{\sqrt{3}|t - t'|}{\ell} \right),
\tag{11}
$$

with magnitude $\tau^2 > 0$ and length-scale $\ell > 0$. We note that to avoid specifying the hyperparameter values, one could estimate them along with the variance components. See Arjas *et al.* (2020) for joint estimation of variance components and curve-specific length-scale parameters. However, we leave this extension for future as it is not the focus of this study.

## Data

We tested the performance of our method with simulated and real data. We used 3 different strategies for simulating data. The first is based on our proposed model and the 2 others on RRM with first- and second-order polynomials. In general, the higher the degree of the polynomial, the more flexible the generated curves are. With all data generation methods, we simulated 100 replicates with different ground truth values to reduce the influence of random fluctuation.

### Simulated data

For the first set of replicates, the ground truth variance functions were drawn randomly from a smooth GP. Precisely, the covariance function of the GP was defined to be Matérn with $\nu = 2.5$, $\ell = 1$, and $\tau^2 = 1$ and the mean function to be 0. We randomly drew covariate values for $n = 1,000$ individuals from uniform distribution between $-1$ and $1$. Uniformly spaced computational grid between the smallest and largest covariate values was constructed and log-variance functions were drawn from the GP at grid points. Diagonal matrices $D_G \in \mathbb{R}^{n \times n}$ and $D_E \in \mathbb{R}^{n \times n}$ were formed by interpolating the log-variance functions from the grid points to the covariate points, exponentiating, taking square roots, and setting the diagonals to those values. The relationship matrix $G$ was formed by simulating $n$ values from standard uniform distribution and computing a matrix $A$ containing pairwise distances between the values. The elements of the relationship matrix were then defined as $G_{i,j} = \exp(-200 A_{i,j})$ for $i, j = 1, \ldots, n$. We then set $K = D_G G D_G + D_E^2$ and generated the data vector $\tilde{y} = r 1_n + Lw$, where $L \in \mathbb{R}^{n \times n}$ is the Cholesky factor of $K$ and $w \in \mathbb{R}^n$ is a standard normal random vector. The scalar $r \in \mathbb{R}$ was drawn from standard uniform distribution to model an intercept term.

The second and third set of replicates were generated by assuming a RRM. In this experiment, we also set the number of individuals $n = 1,000$. The simulated relationship matrix $G$ was generated similarly as in the first set. The genetic and residual random effects were defined by first-order polynomials (lines) for the second set, and second-order polynomials (parabolas) for the third set, in the interval $[-1, 1]$, such that each individual had their own polynomial coefficients. This was achieved by defining a $q$th order ($q$ is either 1 or 2) polynomial basis function matrix $\Phi \in \mathbb{R}^{n \times (q+1)n}$, where elements $i, 1 + (q+1)(i-1) + k$ of the matrix were defined as $\Phi_{i,1+(q+1)(i-1)+k} = t_i^k$, for $k = 0, \ldots, q$, where $t_i$ is the measured covariate value on the ith individual. All other matrix elements were set to zero. The covariate values were again drawn from uniform distribution between $-1$ and $1$. The genetic random effects were then defined as $\tilde{u} = \Phi a$ and residual random effects as $\tilde{e} = \Phi b$, where $a, b \in \mathbb{R}^{(q+1)n}$ are coefficients of the polynomials. Instead of defining the coefficients of the polynomials, we defined their covariance structures separately for the genetic and residual parts as $\Gamma_G = G \otimes L_G L_G^\mathsf{T}$ and $\Gamma_E = I_n \otimes L_E L_E^\mathsf{T}$. Here, $L_G \in \mathbb{R}^{q+1 \times q+1}$ and $L_E \in \mathbb{R}^{q+1 \times q+1}$ are lower triangular matrices whose nonzero elements were simulated from uniform distribution between $-1$ and $1$. Essentially this construction means that the coefficients of the polynomials in the residual part are independent and identically distributed between individuals, with covariance matrix $L_E L_E^\mathsf{T}$ within individuals. For the genetic part, the coefficients between individuals have covariance matrix $G$, and the within-individual covariance matrix is $L_G L_G^\mathsf{T}$. Finally, we defined the data covariance as $K = \Phi \Gamma_G \Phi^\mathsf{T} + \Phi \Gamma_E \Phi^\mathsf{T}$ and set $\tilde{y} = r 1_n + Lw$, where $L$ is the Cholesky factor of $K$ and $w$ is a standard normal random vector. The scalar $r$ was drawn from standard uniform distribution to model an intercept term.

### Real data

The real data set consisted of phenotype and genotype information of heterogeneous stock mice (see Valdar *et al.* 2006 for description of the data set). The data can be downloaded from http://mtweb.cs.ucl.ac.uk/mus/www/GSCAN/. Part of the data also comes with the R package BGLR (Perez 2014). We modeled the genetic and residual variation of high-density lipoprotein (HDL) cholesterol (mg/dL) of mice separately as a function of their age (days) and weight (g). The data were preprocessed by removing all individuals with missing values in either HDL cholesterol, age, or weight. This resulted in a dataset of $n = 1,536$ individuals. In the analyses, we used the pedigree-based additive genetic relationship matrix which came with the R package BGLR.

### Analyses

Simulated data replicates from all 3 simulation models were analyzed with the proposed GP-REBE method as well as first- and second-order polynomial RRM for comparison. In analysis stage, original relationship matrix $G$ used in simulation stage was assumed to be known. In MCMC runs, we used 5,000 iterations and computed the point estimates using posterior mean. For the RRM, we used a reaction norm model which allows modeling of genetic and residual terms as a function of a continuous covariate. The individual genetic and residual terms were modeled as first- and second-order regular polynomials. To achieve this, we used the software MTG2 (Lee and Van der Werf 2016). From the genetic and residual variance estimates $\hat{\sigma}_G^2$ and $\hat{\sigma}_E^2$, respectively, we computed the normalized mean squared error (NMSE) with respect to the ground truths $\sigma_G^2$ and $\sigma_E^2$ as $\mathrm{NMSE}_G = \|\hat{\sigma}_G^2 - \sigma_G^2\|^2 / \|\sigma_G^2\|^2$ and $\mathrm{NMSE}_E = \|\hat{\sigma}_E^2 - \sigma_E^2\|^2 / \|\sigma_E^2\|^2$ for all replicates $i = 1, \ldots, 100$. To mitigate the influence of outliers, we took the median of the NMSEs to form an overall understanding of the quantitative performance of the methods. Running 5,000 iterations of MCMC with $n = 1,000$ and $n = 2,000$ took approximately 5 and 22 min, respectively, on a workstation equipped with Ryzen Threadripper 2950X processor.

The real data sets were analyzed with GP-REBE and the result compared to a standard linear mixed model which assumes constant variances. The standard models were fitted using the function `mixed.solve` in the R package rrBLUP (Endelman 2011).

# Results and discussion
## Simulated data

Figure 2 shows estimation results of GP-REBE and first-order polynomial RRM (RRM (1)) from a single replicate generated with the GP-REBE model with $n = 1,000$. The results from analyzing the data with second-order polynomial RRM are not shown because we could not get the algorithm to converge. The posterior mean follows the ground truth quite well and the 95% credible interval contains the ground truth in the whole covariate domain. The curves estimated with RRM (1) are stiffer than the ones estimated with GP-REBE. This is evident in the residual variance plot where the peak in the left of the domain is not recovered by RRM (1). The amount of uncertainty in the estimates seems to depend on the magnitude of variance: when variance is close to zero, so is the uncertainty. In addition, the uncertainty grows when going toward the boundaries of the domain. This is because nearby measurements share information with each other and there are no measurements outside the boundaries.

The results from analyzing a single data replicate generated with RRM (1) are presented in Fig. 3. Since the genetic and residual

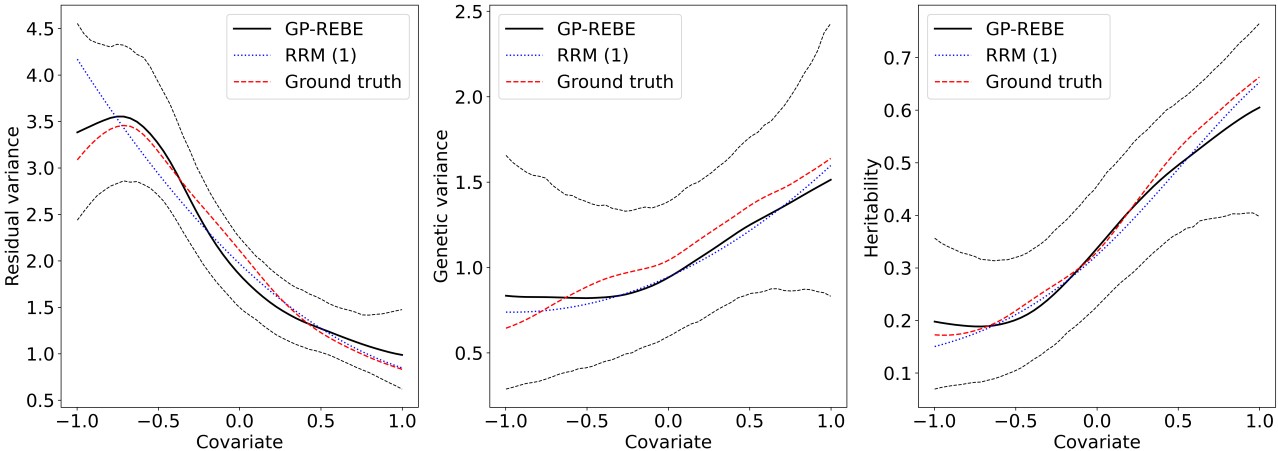

**Fig. 2.** Estimated residual and genetic variance components and narrow-sense heritability from 1 of the replicates generated with the GP-REBE model. The solid black line indicates the posterior mean and dashed black lines the 95% credible interval.

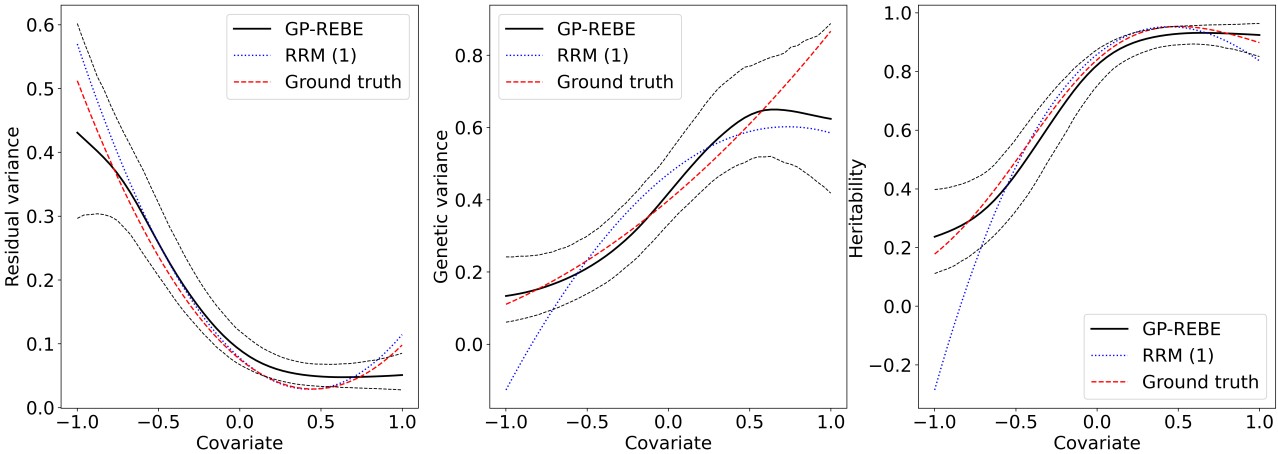

**Fig. 3.** Estimated residual and genetic variance components and narrow-sense heritability from 1 of the replicates generated with first-order polynomial RRM. The solid black line indicates the posterior mean and dashed black lines the 95% credible interval.

random effects are defined as lines, the ground truth variance curves are consequently quadratic. Even though the curves are simple, GP-REBE does a good job recovering them, despite being slightly too flexible. This makes the estimates of RRM (1) slightly more accurate for this kind of data in general.

Finally, Fig. 4 shows the results from analyzing a single data replicate generated with second-order polynomial RRM (RRM (2)). Again, GP-REBE does a decent job in recovering the variance and heritability curves. However, RRM (1) struggles especially with the heritability curve. This results from attempting to approximate quadratic functions with lines.

Additionally, we performed a quantitative comparison of the estimation accuracy of GP-REBE and RRM (1) and RRM (2). The results are shown in Table 1. The residual variance is quite accurately estimated by GP-REBE for all data sets, the error being 1.3–2.7%. The genetic variance is slightly less accurately estimated. In total, GP-REBE performs better than either RRM when analyzing data generated by GP-REBE and RRM (2). When analyzing data generated with RRM (1), RRM (1) as the analysis model performs the best. The poor numbers for RRM (2) as the analysis model are due to failure to converge.

## Real data

The results of analyzing variation of HDL cholesterol with respect to age in mice are presented in Fig. 5. The posterior estimated residual variance seems to decrease with age while the genetic variance increases with age, however the uncertainty is too large to make further conclusions. It is especially large with older (>75 days) mice. This is simply because there are so few mice of that age. The data were also analyzed with standard linear mixed model which assumes constant variances. These are indicated by the dashed blue horizontal lines in the figure. At around age 74, the blue lines exit the 95% credible intervals. This indicates that assuming constant variances is too restrictive, hence dynamic approaches are needed. The results for analyzing these data with RRM are presented in Supplementary Fig. 1.

Finally, Fig. 6 shows the variation in HDL cholesterol in mice with respect to weight. While there is some fluctuation in the estimates, the uncertainty is too high to draw further conclusions. In this case, the constant variance assumptions would not be too violating since the blue horizontal lines are well within the 95% credible intervals. The results for analyzing these data with RRM are presented in Supplementary Fig. 2. The results for real data

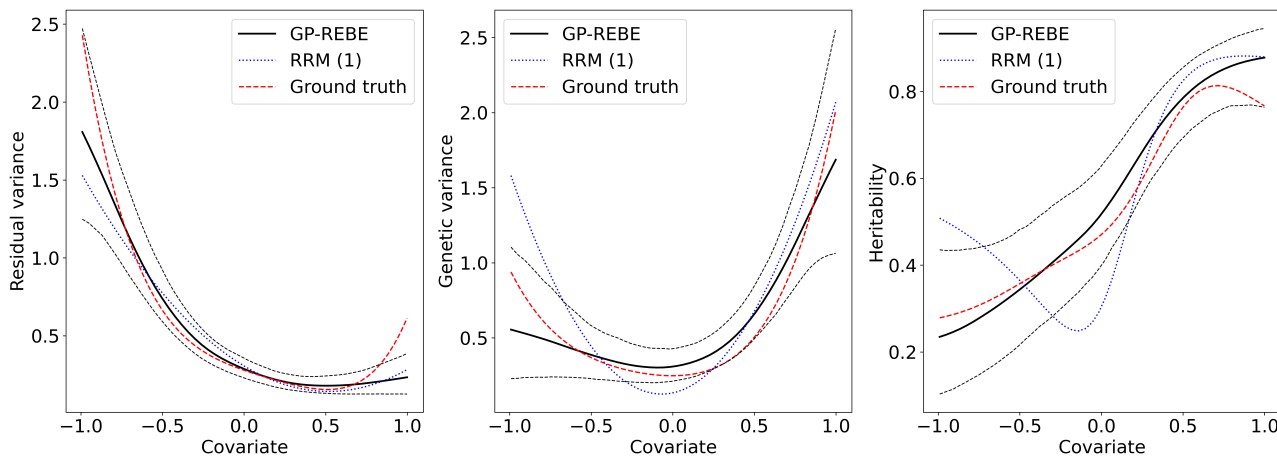

**Fig. 4.** Estimated residual and genetic variance components and narrow-sense heritability from 1 of the replicates generated with second-order polynomial RRM. The solid black line indicates the posterior mean and dashed black lines the 95% credible interval.

**Table 1.** Replicated simulation analyses over 100 data sets.

| | Analysis model | | | | | |
|---|---|---|---|---|---|---|
| | GP-REBE | | RRM (1) | | RRM (2) | |
| Simulation model | Gen. | Res. | Gen. | Res. | Gen. | Res. |
| GP-REBE | **2.8** (1.6) | **1.3** (0.7) | 4.3 (3.6) | 1.6 (1.2) | 41.5 (44.1) | 63.6 (129.6) |
| RRM (1) | 5.3 (4.2) | 1.8 (1.2) | **4.1** (4.9) | **1.0** (1.0) | 37.0 (44.3) | 49.9 (59.6) |
| RRM (2) | **9.3** (5.0) | **2.7** (1.9) | 10.1 (10.7) | 7.6 (8.0) | 40.1 (44.5) | 45.1 (45.5) |

Median (median absolute deviation) of NMSEs of the genetic (Gen.) and residual (Res.) variance curve estimates with respect to the ground truth of our proposed method (GP-REBE) and the RRM with first (RRM (1)) and second (RRM (2)) order polynomials. The bolded values indicate the best performing analysis model for each simulation model.

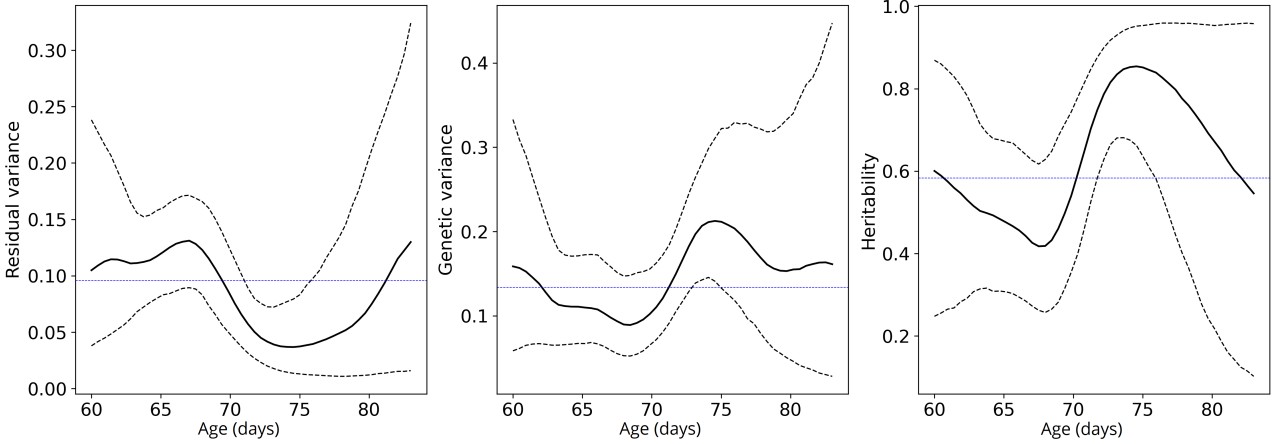

**Fig. 5.** Estimated residual and genetic variance components and narrow-sense heritability of HDL cholesterol of mice as a function of age. The solid black line indicates posterior mean and black dashed lines the 95% credible interval. The blue dashed lines indicate values estimated by a standard REML model assuming constant variances.

analysis are quite different between GP-REBE and RRM. We believe there are multiple reasons for this:

1) Different estimators used (Bayesian vs. REML estimation of RRM). Particularly, the Bayesian estimate is the mean of the posterior distribution, while REML estimate is the maximizer of the likelihood function. If the posterior of the variance components is skewed, these might be radically different.

2) Weak level of relationships between individuals which leads to high uncertainty and skewness in variance component estimates. This is partly related to the first point.

3) Our method maintains positive variance estimates unlike RRM. In Supplementary Fig. 1, the residual variance estimated by RRM goes below zero. This is concerning as we expect all variances to be positive. This is why we believe our results to be more plausible for these data.

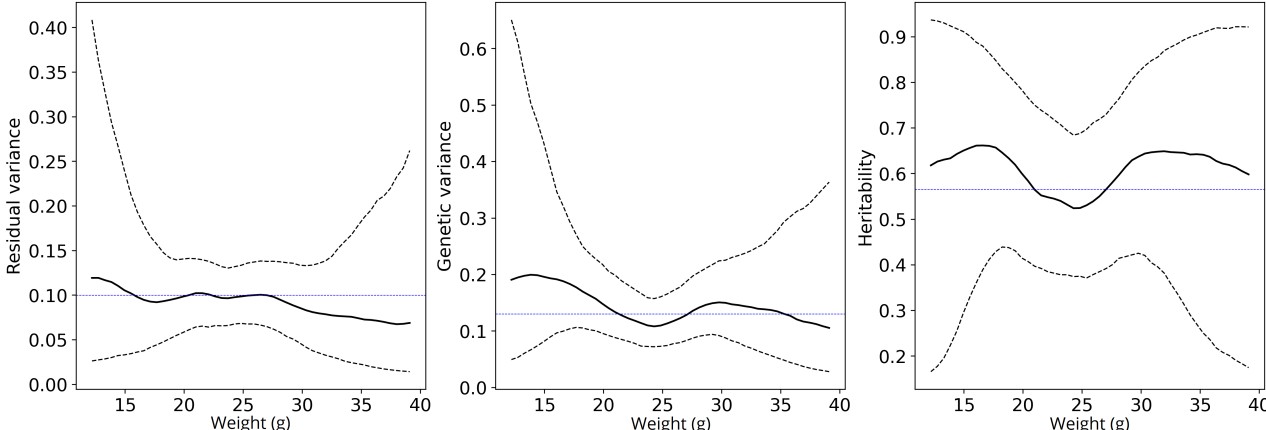

**Fig. 6.** Estimated residual and genetic variance components and narrow-sense heritability of HDL cholesterol of mice as a function of weight. The solid black line indicates posterior mean and black dashed lines the 95% credible interval. The blue dashed lines indicate values estimated by a standard REML model assuming constant variances.

## Discussion

We have developed a novel method GP-REBE for dynamic variance component estimation along a continuous covariate. As the continuous covariate can either be time (group-level longitudinality) or some environmental variable (interaction), the method has a wide variety of potential uses.

In principle, the dynamic linear mixed-effects model introduced here is not limited to 2 random terms $\tilde{u}$ and $\tilde{e}$, just like the usual linear mixed model is not. In practice, considering also dominance, epistasis or including random effects among the environmental systematic effects would be beneficial expansions of the current model in the future.

The assumed form of the covariance matrices of the random terms $\tilde{u}$ and $\tilde{e}$ imposes some compromises. The covariance matrix $D_G G D_G$ of $\tilde{u}$ allows the variance explained additively by the genotypes vary smoothly over time or over the range of the continuous covariate. It is natural however to think that this kind of dependence on the level of explained variance is not simply the result of genetic effects being scaled by the square of the variance component; instead effect sizes of individual genetic variant can change sign (−/0/+) in a complex manner. That is, we would expect the time-dependence to be stronger further away from the diagonal of the covariance matrix, but our numerical experiments with such a model led to identification problems as the covariance structure of $\tilde{u}$ resembled too much that of $\tilde{e}$ (results not shown). Conversely, the covariance matrix $D_E^2$ of $\tilde{e}$ assumes full independence between individuals. It is therefore useful to think of $\tilde{e}$ as the residual instead of "the environment"; the usual issues of confounding by structure still apply. The modeled interaction is limited to take place between genotypes and the single continuous covariate, be it is the time axis or something else.

We compared our method to RRM which is an established method for modeling gene–environment interactions. There are some fundamental differences between how the models are built. First, RRM assumes some parametric functional form for how the genetic and residual terms evolve over the environmental variable, be it discrete or continuous. The functions are usually represented using some known basis and unknown coefficients. The coefficients are different for each individual. Fitting of such a model then amounts to estimating the covariance structure of the coefficients. This is usually done with REML-based approaches. Our method approaches the problem from a different

perspective. We assume that the variance components may take different values depending on the environmental variable. The smoothness of the variance component curves is enforced using a Bayesian approach and placing GP priors on them. The estimation is performed similarly with REML-transformation in the Bayesian framework. There may be favorable use cases for both approaches. In particular, if there is some prior knowledge that the genetic and residual effects are in fact linear functions of the environment, first-order polynomial RRM is a good choice, as indicated by our numerical results. However, if prior knowledge is not available, or if the genetic and residual effects are known to be nonlinear functions, using a more flexible model such as GP-REBE might be more beneficial. In practice, we observed that using higher than first-order polynomial in RRM resulted in unstable behavior in the optimization stage. This could indicate that the objective function has multiple local minima. This behavior was not observed with GP-REBE.

The interpretation of the models is also different. In RRMs, one has to imagine a continuum at the phenotype level since the genetic effect is defined as a continuous function of the environment. In our model, the continuum is at the population (variance) level and there is no need to assume functional relationships at the phenotype level. In our opinion, this is a more straightforward approach if the goal is to estimate variances.

## Data availability

The procedure for generating synthetic data is described in "*Simulated data*" section. The real mouse data are available at http://mtweb.cs.ucl.ac.uk/mus/www/GSCAN/. The GP-REBE software used in this article is publicly available at https://github.com/aarjas/GP-REBE.

Supplemental material available at GENETICS online.

## Acknowledgments

We are grateful to the Editor and 2 anonymous reviewers for their useful comments that helped us to improve the readability of our work. MJS is thankful to Professor Katri Kärkkäinen for discussions concerning the need for this type of methodology especially for forest breeding datasets (Valdar et al. 2006).

## Funding

This work was funded by the University of Oulu and the Research Council of Finland Profi 5 Project 326291 (funding for mathematics and AI: data insight for high-dimensional dynamics). KL was also supported by Research Council of Finland Center of Excellence in Tree Biology (TreeBio AoF CoE 346139).

## Conflicts of interest

The author(s) declare no conflicts of interest.

## Author contributions

AA, KL, and MJS conceptualized the study. AA performed the empirical analyses and drafted the manuscript. All authors interpreted the results, critically revised the manuscript for intellectual content, and approved the submitted version.

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

*Editor: W. Valdar*