## [Peer Review File · Genetics]

Posterior estimation of longitudinal variance components from non-longitudinal data using Bayesian Gaussian process model

Arttu Arjas, Mikko Sillanpää, and Kalle Leppää

NOTE: The reviews and decision letters are unedited and appear as submitted by the reviewers. In extremely rare instances and as determined by a Senior Editor or the EIC, portions of a review may be redacted. If a review is signed, the reviewer has agreed to no longer remain anonymous. The review history appears in chronological order.

Review Timeline:

Submission Date:	2024-04-30
Editorial Decision:	2024-07-07
Resubmission Received:	2025-01-09
Accepted:	2025-02-14

July 7, 2024

GENETICS-2024-307056

Posterior estimation of longitudinal variance components from non-longitudinal data using Bayesian Gaussian process model

Dear Dr. Arjas:

Two researchers in the field have reviewed your manuscript, and I have read it as well. The described extension of the longitudinal GP model to non-longitudinal data is potentially useful and may have wide application. However, the manuscript falls short in several areas, including adequate description and explanation of comparison methods and some questions about the extent and rigor of the simulations. Moreover, the superiority of the GP model is not obvious in the results that average MSE over multiple simulations. As such, your manuscript is not currently acceptable for publication in GENETICS. However, we would welcome a substantially revised manuscript for reconsideration. Both the reviewers and myself have comments and concerns to be addressed in a revised manuscript. You can read their and my reviews at the end of this email.

Upon resubmission, please include:

1. A clean version of your manuscript;
2. A marked version of your manuscript in which you highlight significant revisions carried out in response to the major points raised by the editor/reviewers (track changes is acceptable if preferred);
3. A detailed response to the editor's/reviewers' feedback and to the concerns listed above. Please reference line numbers in this response to aid the editor and reviewers.

Your paper will be sent back out for review.

Additionally, please ensure that your resubmission is formatted for GENETICS
<https://academic.oup.com/genetics/pages/general-instructions>

Follow this link to submit the revised manuscript: Link Not Available

Sincerely,

William Valdar
Associate Editor
GENETICS

Approved by:
Hongyu Zhao
Senior Editor
GENETICS

Reviewer #1 (Comments for the Authors (Required)):

In this manuscript, the authors propose a "dynamic" linear mixed model to handle datasets where each individual contributes only a single phenotypic measurement, but the variance components vary dynamically with a continuous covariate like time points. Compared to the mixed model with static genetic and residual variance-covariance matrices calculated across time points, the genetic and residual variance-covariance matrices in the dynamic model are scaled differently according to the time point (i.e., "dynamic variance components"). Specifically, they introduced $\sigma_G(t_i)$ and $\sigma_E(t_i)$, which form the diagonal matrices D_G and D_E . Note that D_G and D_E alone do not determine the genetic and residual variance-covariance matrices. Instead, they scale the genetic and residual variance-covariance matrices according to the time point t_i . In detail, the posterior distributions of $\sigma_G(t_i)$ and $\sigma_E(t_i)$ were estimated using a Bayesian Gaussian Process model, and a Metropolis-adjusted Langevin algorithm was used for efficient posterior sampling.

Although a random regression model (RRM) can also handle such datasets to estimate time-point-specific variances, RRM may be unstable when including more than intercept and slope parameters. The authors used both simulation and real datasets to demonstrate that their method had similar or better performance than RRM for variance components estimation at each time point. The overall writing quality is high, and the logic is smooth. As I continued reading, many questions were answered.

Comments:

Page 3, section 2.1.2: Since D_G and D_E are diagonal matrices, should the time points be assumed independent? However, the authors mentioned, "In order to model dependence between time points." My understanding is that even though D_G is diagonal and suggests independence across different time points for a given individual, the covariance between the random effects (u_{tilda}) across individuals at potentially the same or different time points is influenced by G . Why is there a Matern covariance in section 2.1.3 when we assume time points are independent?

Minor comments:

Page 3, section 2.1.2: The dimensions in equation (2) are missing. Assuming $Z=I$, are the diagonal matrices D_G and D_E of dimension n -by- n ? If so, do individuals at the same time point have the same diagonal entries in D_G and D_E ?

Reviewer #2 (Comments for the Authors (Required)):

The authors developed a novel method to estimate variance components from longitudinal data which is applicable to situations where phenotypic values are taken only once from a genotype. The method is interesting and statistically sound. But it is not fully illustrated when the proposed method is advantageous to existing ones including RRM and Arjas et al. (2020).

Major comments

The simulations are too concise. The authors should characterize the proposed method with more details. When and how is the proposed method superior to the existing methods? Is it related to the number of phenotypic records per genotype, sparseness of the records, heritability, and covariances across points? Is it related to estimation accuracy or computational efficiency such as convergence?

The discussion of the real data analysis is also too concise. The results obtained by the proposed method and RRM are quite different, but no discussion is provided. Which is more plausible? And it would be better to use multiple real data.

Other comments

The compared method (RRM) is not described well (4.3 Analyses). Please explain such that readers can reproduce.

Section 2.1.2: Whereas the dimension of G is the number of genotypes, the dimension of D_G is the number of points. So the product $D_G * G * D_G$ seems to be uncomfortable. Also, the indices for points i and j are used for G (G_{ij}). Is this correct?

Section 3.1: Aren't the points where phenotypes are sampled included in the grids? Is this not influential on inference accuracy?

Section 3.3: It will be preferred to provide preliminary results about hyperparameter specification.

Associate Editor Comments:

Please put sections into standard Genetics format, ie, sections 2 ("Methods"), 3 ("Implementation"), and 4 ("Data"), should all be in a single Methods section with suitable subheadings.

The manuscript compares the performance of GP-REBE with RRM, but nowhere are RRM described explicitly so that readers unfamiliar with that specific model would have to look elsewhere. Please add a subsection to Methods explicitly describing the RRM(s) that is being compared. Also move Figure 5 to the Methods, so that readers can use the diagram to accompany the mathematical descriptions.

p6 In hyperparameters of the GP, help the reader by writing out the Bessel function for $\nu=1.5$.

Please state explicitly in the Methods how the GP-REBE differs from the Arjas et al, 2020. This is alluded to in the Introduction, but since the difference is crucial to the novelty of the manuscript, it is essential that it is explicitly stated.

p8-9 A major point of the paper is demonstrate the behavior of the GP model vs the RRM, but having RRM results in Supplement hinders this comparison. Please add incorporate the RRM results into Figures 1 and 2.

The RRM results Supplementary Figure 1 demonstrate quadratic fits, suggesting the use of a 2nd order polynomial. However, in the very terse description of the RRM used in sec 4.3, you state the RRM use first order polynomials --- is this a mistake?

The simulated data from the GP are more nuanced that could be captured by a linear or quadratic term. I understand that this is the point of the manuscript, however, I think it would be helpful to also show results from higher order polynomial RRM if possible. I realize that you mention that higher order polynomials are less stable, but it seems likely that you could go further than the simple RRM used in the current manuscript.

Would the instability of the higher order polynomial RRM be resolved if they were fitted using a Bayesian method (in which

case, multimodality would not be a problem)?

Specifically, whether or not higher order polynomial RRM's are implemented, it would be useful to see some formal description of the trade-off between model complexity and fit and compare this with the constant complexity of the GP model.

I am confused by the description "50 simulation replicates" (eg, "Analyses" on p8). I presume this means 50 complete simulated datasets, with 50 different trajectories (ie, 50 different covariate-variance curves)? (If not, then this is a problem.) If so, then I do not understand how Figure 1 can represent all of them, as suggested by the Figure legend ("Estimated... from 50 simulated data set replicates"), since it looks like you are showing the analysis of one example simulated data from the 50 available. If is showing one simulation, then that is fine, but please clarify and put results from the remaining 49 in the supplement, either as a full set of graphs for each or as true and estimated heritability for each, as it will be useful for readers to see the variety of curves examined.

Why are there only 50 simulations? Often in statistical methods papers, the number of simulated data sets is greater, eg, 1000. If this is because of computation time, then say so. Please also anyway provide details on computation speed, as this will be a concern for potential users.

Table 1 summarizes averaged related MSE from the analysis of the 50 + 50 = 100 simulations. Since these are averages, please add standard errors or confidence intervals so the numbers can be compared meaningfully.

Table 1 could be clearer. For example, make the first (currently unnamed) column "Simulation model", and replace Repl. 1 and Repl. 2 with GP-REBE and RRM respectively (if that is what is meant).

The results in Table 1 seem to show that GP-REBE and RRM are similar, without a clear advantage of the proposed GP model. I should note here that it seems like the flexibility of the GP model means it could model trajectories simulated from very high order polynomial RRM's, including those that could not be easily analyzed using the RRM's of the correct polynomial order. This could be a useful way to demonstrate superiority?

The documentation in the github repo provided (which in its entirety comprises the text "Estimation of longitudinal variance components") is not adequate. Please include scripts to run all the analyses in the manuscript and suitable documentation, including vignettes if possible, so that others can reproduce your analysis and, ideally, apply GP-REBE to their data.

Minor:

p5 "produce of" -> "produce the"

The Mrode reference in the Bibliography is misformed.

Detailed response to reviewers' comments

Reviewer 1

Comment: Although a random regression model (RRM) can also handle such datasets to estimate time-point-specific variances, RRM may be unstable when including more than intercept and slope parameters. The authors used both simulation and real datasets to demonstrate that their method had similar or better performance than RRM for variance components estimation at each time point. The overall writing quality is high, and the logic is smooth. As I continued reading, many questions were answered.

Our response: We thank the Reviewer for the kind comments.

Comment: Page 3, section 2.1.2: Since D_G and D_E are diagonal matrices, should the time points be assumed independent? However, the authors mentioned, "In order to model dependence between time points.". My understanding is that even though D_G is diagonal and suggests independence across different time points for a given individual, the covariance between the random effects \tilde{u} across individuals at potentially the same or different time points is influenced by G . Why is there a Matern covariance in section 2.1.3 when we assume time points are independent?

Our response: We do not assume that time points are independent. The matrices D_E and D_G are diagonal just to collect the variance values at each measured time point. This is required to specify the model. Even though the matrices are diagonal, we may enforce some smoothness in the variance curves through the Matern covariance which models the dependencies between each time point. We have clarified this in the paper.

Comment: Page 3, section 2.1.2: The dimensions in equation (2) are missing. Assuming $Z = I$, are the diagonal matrices D_G and D_E of dimension n -by- n ? If so, do individuals at the same time point have the same diagonal entries in D_G and D_E ?

Our response: Yes, D_G and D_E are of size $n \times n$, and two individuals with the same measured time point have the same diagonal entries for D_G and D_E . We have added the sentence "The dimensions of the variables are the same as in Eq. (1)" below Eq. (2) to clarify the dimensions.

Reviewer 2

Comment: The simulations are too concise. The authors should characterize the proposed method with more details. When and how is the proposed method superior to the existing methods? Is it related to the number of phenotypic records per genotype, sparseness of the records, heritability, and covariances across points? Is it related to estimation accuracy or computational efficiency such as convergence?

Our response: We have expanded the simulations so that we have now three different simulation models and also three different analysis models: GP-REBE and RRM with first order polynomials and RRM with second order polynomials. All simulated datasets have been analyzed with all three analysis models. This is to show which model may be best suited to which scenario. The conclusion is that in majority of cases, GP-REBE showed the best performance. The exception was when analyzing data generated with first-order polynomial RRM with the same analysis model. To conclude, we believe GP-REBE is best suited for analyzing higher than first order functional shapes for genetic values. This is clearly indicated by the numerical results in Table 1. Moreover, in this regime, the convergence properties of GP-REBE are better than those of higher order polynomial RRMs. We have included these thoughts in Section 3.2.

Comment: The discussion of the real data analysis is also too concise. The results obtained by the proposed method and RRM are quite different, but no discussion is provided. Which is more plausible? And it would be better to use multiple real data.

Our Response: We believe the difference in real data analysis has multiple reasons:

1. Different estimators used (Bayesian vs. REML estimation of RRM). Particularly, the Bayesian estimate is the mean of the posterior distribution, while REML estimate is the maximizer of the likelihood function. If the posterior of the variance components is skewed, these might be radically different.
2. Weak level of relationships between individuals which leads to high uncertainty and skewness in variance component estimates. This is partly related to the first point.
3. Our method maintains positive variance estimates unlike REML estimation for RRM. In Supplementary Figure 1, the residual variance estimated by RRM goes below zero. This is concerning as we expect variances to be positive. This is why we believe our results to be more plausible for this data.

We have included these considerations in the Results and discussion section.

Comment: The compared method (RRM) is not described well (4.3 Analyses). Please explain such that readers can reproduce.

Our response: We have added a subsection 2.3 “Random regression models” with a brief description of RRM and how they are fitted.

Comment: Section 2.1.2: Whereas the dimension of G is the number of genotypes, the dimension of D_G is the number of points. So the product $D_G G D_G$ seems to be uncomfortable. Also, the indices for points i and j are used for $G(G_{i,j})$. Is this correct?

Our response: The dimension of G is $n \times n$, the same as the number of individuals. We assume that each individual i has a measured time point (or other continuous covariate), say t_i . The matrices D_E and D_G consist of the square root of the variance function values at the measured time points, i.e., $D_E = \text{diag}(\sigma_E(t_1), \dots, \sigma_E(t_n))$, $D_G = \text{diag}(\sigma_G(t_1), \dots, \sigma_G(t_n))$. This makes D_E and D_G $n \times n$ matrices as well. The notation G_{ij} means the genomic coefficients of relationship between individual i and j . We have added the dimensions of G , D_E and D_G to section 2.1.1.

Comment: Section 3.1: Aren't the points where phenotypes are sampled included in the grids? Is this not influential on inference accuracy?

Our response: This is an interesting comment and likely true. In the current version of the algorithm, the points where phenotypes are sampled (measured time points) are not included in the grids. Interpolating with the Gaussian process from the grid to the measured time points introduces some additional uncertainty, depending on the distance of the grid points from the measured time points. Including the measured time points in the grid would eliminate the problem. However, it would increase the number of unknowns in the model by $2n$ and cause other issues that way. Moreover, if there are no large 'gaps' in the measured time points, we would expect the loss of accuracy to be quite small. We have added some discussion on this to Section 3.1.

Comment: Section 3.3: It will be preferred to provide preliminary results about hyperparameter specification.

Our comment: We added short comment on specification of ℓ in section 2.4.3.

Associate editor

Comment: Please put sections into standard Genetics format, ie, sections 2 (“Methods”), 3 (“Implementation”), and 4 (“Data”), should all be in a single Methods section with suitable subheadings.

Our response: The sections are now in standard Genetics format.

Comment: The manuscript compares the performance of GP-REBE with RRM, but nowhere are RRM described explicitly so that readers unfamiliar with that specific model would have to look elsewhere. Please add a subsection to Methods explicitly describing the RRM(s) that is being compared. Also move Figure 5 to the Methods, so that readers can use the diagram to accompany the

mathematical descriptions.

Our response: We have added section 2.3 “Random regression models” with a brief description of RRM and how they are fitted. We also moved Figure 5 to the Methods section.

Comment: p6 In hyperparameters of the GP, help the reader by writing out the Bessel function for $\nu = 1.5$.

Our response: We added the Matérn covariance function for $\nu = 1.5$.

Comment: Please state explicitly in the Methods how the GP-REBE differs from the Arjas et al, 2020. This is alluded to in the Introduction, but since the difference is crucial to the novelty of the manuscript, it is essential that it is explicitly stated.

Our response: We added the following statement at the end of section 2.1.2: “Moreover, we note that this model is an extension of the model presented in Arjas et al. [2020]. The previous work assumed that all n individuals were measured at t common, equidistant time points, yielding a total of nt measurements. In this work we assume that there are in total n measurements taken at arbitrary time/covariate points.”

Comment: p8-9 A major point of the paper is demonstrate the behavior of the GP model vs the RRM, but having RRM results in Supplement hinders this comparison. Please add incorporate the RRM results into Figures 1 and 2.

Our response: We have incorporated RRM results in the figures.

Comment: The RRM results Supplementary Figure 1 demonstrate quadratic fits, suggesting the use of a 2nd order polynomial. However, in the very terse description of the RRM used in sec 4.3, you state the RRM use first order polynomials — is this a mistake?

Our response: This is not a mistake. When fitting a RRM with first order polynomials, the resulting variance curves will be of second order, resulting from computing the variances as diagonal values of $\Phi C \Phi^T$, where Φ contains the basis functions the C is the covariance matrix of the RRM coefficients. In general, the variance curves will always be of higher order than the polynomial used to model the random effects.

Comment: The simulated data from the GP are more nuanced that could be captured by a linear or quadratic term. I understand that this is the point of the manuscript, however, I think it would be helpful to also show results from higher order polynomial RRM if possible. I realize that you mention that higher order polynomials are less stable, but it seems likely that you could go further than the simple RRM used in the current manuscript.

Our response: We have now included RRM second-order polynomials as simulation and analysis models in the comparison.

Comment: Would the instability of the higher order polynomial RRM be

resolved if they were fitted using a Bayesian method (in which case, multimodality would not be a problem)?

Our response: To some degree, maybe. However, using Markov Chain Monte Carlo methods efficiently with multimodal distributions is also a nontrivial task.

Comment: Specifically, whether or not higher order polynomial RRM's are implemented, it would be useful to see some formal description of the trade-off between model complexity and fit and compare this with the constant complexity of the GP model.

Our response: We have now included comparison with first and second-order polynomial RRM's. Even though higher order polynomial RRM's should in theory increase estimation accuracy, fitting of such models in practice is very inconsistent. This is clearly indicated by the results in Table 1 as very high MSE numbers for second-order polynomial RRM's.

Comment: I am confused by the description "50 simulation replicates" (eg, "Analyses" on p8). I presume this means 50 complete simulated datasets, with 50 different trajectories (ie, 50 different covariate-variance curves)? (If not, then this is a problem.) If so, then I do not understand how Figure 1 can represent all of them, as suggested by the Figure legend ("Estimated... from 50 simulated data set replicates"), since it looks like you are showing the analysis of one example simulated data from the 50 available. If is showing one simulation, then that is fine, but please clarify and put results from the remaining 49 in the supplement, either as a full set of graphs for each or as true and estimated heritability for each, as it will be useful for readers to see the variety of curves examined.

Our response: Our initial idea was to generate 50 replicate data sets with the same ground truth variance curves. This way, only the 'random' part of the data would change and the systematic part would stay the same. However, as you suggested, we have replaced this with 100 data sets with different ground truth curves, of which we show one in the manuscript.

Comment: Why are there only 50 simulations? Often in statistical methods papers, the number of simulated data sets is greater, eg, 1000. If this is because of computation time, then say so. Please also anyway provide details on computation speed, as this will be a concern for potential users.

Our response: It is indeed about computation time. It takes about 5 minutes to analyze one data set with $n = 1000$. We increased the number of simulations to 100. We added some details on the computation speed in section 2.5.3.

Comment: Table 1 summarizes averaged related MSE from the analysis of the $50 + 50 = 100$ simulations. Since these are averages, please add standard errors or confidence intervals so the numbers can be compared meaningfully.

Our response: We have added standard errors to the values.

Comment: Table 1 could be clearer. For example, make the first (currently

unnamed) column "Simulation model", and replace Repl. 1 and Repl. 2 with GP-REBE and RRM respectively (if that is what is meant).

Our response: We have modified Table 1 as suggested.

Comment: The results in Table 1 seem to show that GP-REBE and RRM are similar, without a clear advantage of the proposed GP model. I should note here that it seems like the flexibility of the GP model means it could model trajectories simulated from very high order polynomial RRM, including those that could not be easily analyzed using the RRM of the correct polynomial order. This could be useful way to demonstrate superiority?

Our response: We used this idea in our experimental validation. We now have three different simulation and analysis models: GP-REBE, and first and second order polynomial RRM. All datasets generated by these methods were also analyzed by each method. GP-REBE performed the best when analyzing data generated with GP-REBE and second order polynomial RRM. Since estimation of variances is quite inaccurate in general, even more complex trajectories would require very large datasets to reliably recover.

Comment: The documentation in the github repo provided (which in its entirety comprises the text "Estimation of longitudinal variance components") is not adequate. Please include scripts to run all the analyses in the manuscript and suitable documentation, including vignettes if possible, so that others can reproduce your analysis and, ideally, apply GP-REBE to their data.

Our response: We have added details to the GitHub repository to help users reproduce our results. The comments in the file run.py should contain all necessary details to run and understand the code.

Comment: p5 "produce of" → "produce the".

Our response: We corrected this.

Comment: The Mrode reference in the Bibliography is misformed.

Our response: This is a book and thus the reference looks different.

February 14, 2025

RE: GENETICS-2025-307782

Dr. Arttu Arjas
Oulun yliopisto
Centre for Wireless Communication
Pentti Kaiteran katu 1
Oulu, N/A 90570
Finland

Dear Dr. Arjas:

Congratulations! We are delighted to inform you that your manuscript titled "Posterior estimation of longitudinal variance components from non-longitudinal data using Bayesian Gaussian process model" is acceptable for publication in GENETICS. Many thanks for submitting your research to the journal.

The reviewers had a few suggestions for improving the manuscript that you may want to consider. These are optional. You can view their comments at the bottom of this email.

To Proceed to Production:

1. Format your article according to GENETICS style, as discussed at <https://academic.oup.com/genetics/pages/general-instructions>, and upload your final files at <https://genetics.msubmit.net>.
2. Your manuscript will be published as-is (unedited-as submitted, reviewed, and accepted) at the GENETICS website as an Advanced Access article and deposited into PubMed shortly after receipt of source files and the completed license to publish. Please notify sourcefiles@thegsajournals.org if you do not wish to publish your article via Advanced Access.
3. We invite you to submit an original color figure related to your paper for consideration as cover art. Please email your submission to the editorial office or upload it with your final files. You can submit a small-sized image for evaluation, and if selected, the final image must be a TIFF file 2513px wide by 3263px high (8.375 by 10.875 inches; resolution of 600ppi). Please avoid graphs and small type.

If you have any questions or encounter any problems while uploading your accepted manuscript files, please email the editorial office at sourcefiles@thegsajournals.org.

Sincerely,

William Valdar
Associate Editor
GENETICS

Approved by:
Hongyu Zhao
Senior Editor
GENETICS

Reviewer #1 :

The previous comments have been well addressed. Below are some minor comments.

1. Figure 1: For GP-REBE, consider changing σ^2_G to $\sigma^2_{G(ti)}$ to be clearer.
2. Section 2.3 RRM, line 207: Consider adding that it is assumed that each individual has only one observation. This is because RRM can handle scenarios where different individuals have different numbers of observations.
3. Section 2.5.2 Real Data: Describe the dimensions of the data.
4. Could you provide a rough computational cost of your method and RRM?
5. Instead of using REML as the solver, the RRM has been extended into the Bayesian framework recently (<https://doi.org/10.1002/tpg2.20228>), maybe this can be another benchmark in data analysis.

6. Can the software handle the scenario when some individuals have multiple observations? Or it only allow each individual to have one observation?

Reviewer #2 :

The manuscript has been adequately revised. I have no further comments.